# CoFiNet: Reliable Coarse-to-fine Correspondences for Robust Point Cloud Registration

**Hao Yu**[1,2]    **Fu Li**[1,2]    **Mahdi Saleh**[1]    **Benjamin Busam**[1]    **Slobodan Ilic**[1,3]

[1] Technical University of Munich    [2] National University of Defense Technology
[3] Siemens AG, München
{hao.yu, fu.li, m.saleh, b.busam, slobodan.ilic}@tum.de

## Abstract

We study the problem of extracting correspondences between a pair of point clouds for registration. For correspondence retrieval, existing works benefit from matching sparse keypoints detected from dense points but usually struggle to guarantee their repeatability. To address this issue, we present CoFiNet - **Co**arse-to-**Fi**ne **Net**work which extracts hierarchical correspondences from coarse to fine without keypoint detection. On a coarse scale and guided by a weighting scheme, our model firstly learns to match down-sampled nodes whose vicinity points share more overlap, which significantly shrinks the search space of a consecutive stage. On a finer scale, node proposals are consecutively expanded to patches that consist of groups of points together with associated descriptors. Point correspondences are then refined from the overlap areas of corresponding patches, by a density-adaptive matching module capable to deal with varying point density. Extensive evaluation of CoFiNet on both indoor and outdoor standard benchmarks shows our superiority over existing methods. Especially on 3DLoMatch where point clouds share less overlap, CoFiNet significantly outperforms state-of-the-art approaches by at least 5% on *Registration Recall*, with at most two-third of their parameters. [Code]

## 1  Introduction

Correspondence search is a core topic of computer vision and establishing reliable correspondences is a key to success in many fundamental vision tasks, such as tracking, reconstruction, flow estimation, and particularly, point cloud registration. Point cloud registration aims at recovering the transformation between a pair of partially overlapped point clouds. It is a fundamental task in a wide range of real applications, including scene reconstruction, autonomous driving, simultaneous localization and mapping (SLAM), etc. However, due to the unordered and irregular properties of point clouds, extracting reliable correspondences from them has been a challenging task for a long time. From early-stage hand-crafted methods [1, 2, 3, 4, 5] to recently emerged deep learning-based approaches [6, 7, 8, 9, 10, 11, 12], many works contributed to improving the reliability of correspondences.

We can broadly categorize recent deep learning-based point cloud registration methods into three categories. The first [13, 14, 15] follows the idea of ICP [16], where they iteratively find dense correspondences and compute pose estimation. The second [17, 18] includes the correspondence-free methods based on the intuition that the feature distance between two well-aligned point clouds should be small. Such methods encode the whole point cloud as a single feature and iteratively optimize the relative pose between two frames by minimizing the distance of corresponding features. Though achieving reasonable results on synthetic object datasets [19], both of them struggle on large-scale real benchmarks [20, 6], as the first suffers from low correspondence precision and high computational complexity, while the second lacks robustness to noise and partial overlap.

35th Conference on Neural Information Processing Systems (NeurIPS 2021).

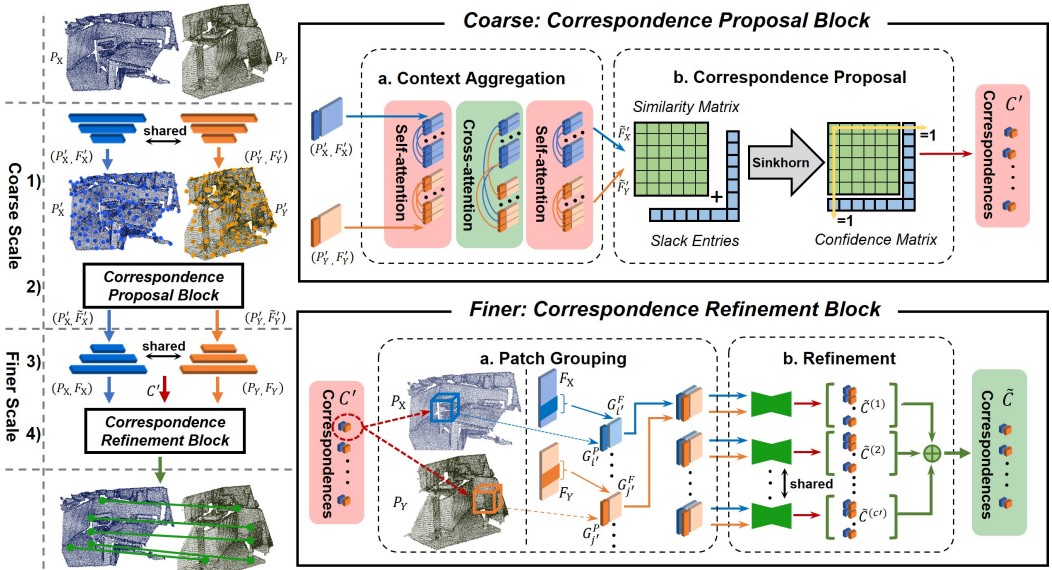

Figure 1: **Left**: Overview of CoFiNet. From top to bottom: 1) Dense points are down-sampled to uniformly distributed nodes, while associated features are jointly learned. 2) Correspondence Proposal Block (**Top Right**): Features are strengthened and used to calculate the similarity matrix. Coarse node correspondences are then proposed from the confidence matrix. 3) Strengthened features are decoded to dense descriptors associated with each input point. 4) Correspondence Refinement Block (**Bottom Right**): Coarse node proposals are first expanded to patches via grouping. Patch correspondences are then refined to point level by our proposed density-adaptive matching module, whose details can be found in Fig. 2.

Differently, the last category of methods [6, 7, 8, 9, 10, 11, 12] tackles point cloud registration in a two-stage manner. They firstly learn local descriptors of down-sampled sparse points (nodes) for matching, and afterward use robust pose estimators, e.g., RANSAC [21], for recovering the relative transformation. Their two-stage strategy makes them achieve state-of-the-art performance on large-scale real benchmarks [20, 6]. Uniform sampling [7, 8] and keypoint detection [22, 10, 11, 12] are two common ways to introduce sparsity. Compared to uniform sampling that samples points randomly, keypoint detection estimates the saliency of points and samples points with strong geometry features, which significantly reduces the ambiguity of matching. However, the sparsity by nature challenges repeatability, i.e., sub-sampling increases the risk where a certain point loses its corresponding point in the other frame, which constrains the performance of detection-based methods [22, 10, 11, 12].

Recently, a coarse-to-fine mechanism is leveraged by our 2D counterparts [23, 24, 25] to avoid direct keypoint detection, which shows superiority over the state-of-the-art detection-based method [26]. However, in 3D point cloud matching, where keypoint detectors usually perform worse, existing deep learning-based methods do not yet exploit such a coarse-to-fine strategy. To fill the gap, we focus on leveraging the coarse-to-fine mechanism to eliminate the side effects of detecting sparse keypoints.

Nevertheless, designing such a coarse-to-fine pipeline in point cloud matching is non-trivial, mainly due to the inherent unordered and irregular nature of point clouds. To this end, we propose a weighting scheme for coarse node matching and a density-adaptive matching module for correspondence refinement, which enables CoFiNet to extract coarse-to-fine correspondences from point clouds. More specifically, on a coarse scale, the weighting scheme proportional to local overlap ratios guides the model to propose correspondences of nodes whose vicinity areas share more overlap, which effectively squeezes the search space of the consecutive refinement. On a finer scale, the density-adaptive matching module refines coarse correspondence proposals to point level by solving a differentiable optimal transport problem [26] with awareness to varying point density, which shows more robustness on irregular points. An overview of our proposed method can be found in Fig. 1. Our main contributions are summarized as follows:

- A detection-free learning framework that treats point cloud registration as a coarse-to-fine correspondence problem, where point correspondences are consecutively refined from coarse proposals that are extracted from unordered and irregular point clouds.

- A weighting scheme that, on a coarse scale, guides our model to learn to match uniformly down-sampled nodes whose vicinity areas share more overlap, which significantly shrinks the search space for the refinement.
- A differentiable density-adaptive matching module that refines coarse correspondences to point level based on solving an optimal transport problem with awareness to point density, which is more robust to the varying point density.

To the best of our knowledge, we are the first deep learning-based work that incorporates a coarse-to-fine mechanism in correspondence search for point cloud registration. Extensive experiments are conducted on both indoor and outdoor benchmarks to show our superiority. Notably, CoFiNet surpasses the state-of-the-art with much fewer parameters. Compared to [12], we only use around two-third and one-fourth of parameters on indoor and outdoor benchmarks, respectively.

## 2 Related Work

**Learned local descriptors.** Early networks proposed to learn local descriptors for 3D correspondence search mainly take uniformly distributed local patches as input. As a pioneer, Zeng et al. [6] propose the 3DMatch benchmark, on which they exploit a Siamese network [27] that consumes voxel grids of TDFs (Truncated Distance Fields) to match local patches. PPFNet [7] directly consumes raw points augmented with point-pair features (PPF) by leveraging PointNet [28] as its encoder. PPF-FoldNet [8] leverages only PPF, which is naturally rotation-invariant, as its input and further incorporates a FoldingNet [29] architecture to enable the unsupervised training of rotation-invariant descriptors. Gojcic et al. [9] propose a network to consume the smoothed density value (SDV) representation aligned to the local reference frame (LRF) to eliminate the rotation-variance of learned descriptors. To extract better geometrical features, Graphite [11] utilizes graph neural networks for local patch description. SpinNet [30] utilizes LRF for patch alignment and 3D cylindrical convolution layers for feature extraction, achieving the best generalization ability to unseen datasets. However, patch-based methods usually suffer from low computational efficiency, as typically shared activations of adjacent patches are not reused. To address this, FCGF [31] makes the first attempt by using sparse convolutions [32] to compute dense descriptors of the whole point cloud in a single pass, which leads to 600x speed-up while still being able to achieve comparable performance to patch-based methods.

**Learned 3D keypoint detectors.** USIP [22] learns to regress the position of the most salient point in each local patch in a self-supervised manner. However, it suffers from degenerated cases when the number of desired keypoints is relatively small. D3Feat [10] exploits a fully convolutional encoder-decoder architecture for joint dense detection and description. However, it does not consider overlap relationships and shows low robustness on low-overlap scenarios. In addition to jointly estimating salient scores and learning local descriptors, PREDATOR [12] also predicts dense overlap scores that indicate the confidence whether points are on the overlap regions. Keypoints will be sampled under the condition of both saliency and overlap scores. Though it surpasses existing methods by a large margin on both 3DMatch[6] and 3DLoMatch[12], the precision of estimated scores and the repeatability of sampled keypoints constrain its performance.

**Coarse-to-fine correspondences.** As witnessed in 2D image matching, many recent works [23, 24, 25] leverage a coarse-to-fine mechanism to eliminate the inherent repeatability problem in keypoint detection and thus boost the performance. DRC-Net [23] utilizes 4D cost volumes to enumerate all the possible matches and establishes pixel correspondences in a coarse-to-fine manner. Concurrently with DRC-Net, Patch2Pix [24] first establishes patch correspondences and then regresses pixel correspondences according to matched patches. In a similar coarse-to-fine manner with Patch2Pixel, LoFTR [25] leverages Transformers [33], together with an optimal transport matching layer [26], to match mutual-nearest patches on the coarse level, and then refines the corresponding pixel of the patch center on the finer level.

## 3 Methodology

**Problem statement.** Given a pair of unordered point sets $\mathbf{X}$ with $n$ points $x_i \in \mathbb{R}^3$ and $\mathbf{Y}$ with $m$ points $y_j \in \mathbb{R}^3$, we aim at recovering the rigid transformation $\overline{\mathbf{T}}_{\mathbf{Y}}^{\mathbf{X}} \in SE(3)$ between them. For simpler notation, we define their coordinate matrices as $\mathbf{P_X} \in \mathbb{R}^{n \times 3}$ and $\mathbf{P_Y} \in \mathbb{R}^{m \times 3}$, respectively. We follow the path of extracting correspondences first and then estimating the relative pose, where we mainly focus on the former. For this purpose, we propose CoFiNet that takes a pair of point clouds as

input and outputs point correspondences, which can be leveraged to estimate the rigid transformation by RANSAC [21].

## 3.1 Coarse-scale Matching

**Point encoding.** On the coarse level, our target is matching uniformly down-sampled nodes whose vicinity areas share more overlap. To achieve this goal, we first adopt shared KPConv [34] encoders to down-sample raw points to uniformly distributed nodes $\mathbf{P}'_{\mathbf{X}} \in \mathbb{R}^{n' \times 3}$ and $\mathbf{P}'_{\mathbf{Y}} \in \mathbb{R}^{m' \times 3}$, while jointly learning their associated features $\mathbf{F}'_{\mathbf{X}} \in \mathbb{R}^{n' \times b}$ and $\mathbf{F}'_{\mathbf{Y}} \in \mathbb{R}^{m' \times b}$. Demonstration of down-sampled nodes can be found in **1)** of Fig. 1. Please refer to Appendix for more details of network architecture.

**Attentional feature aggregation.** As illustrated in Fig. 1, the Correspondence Proposal Block (CPB) takes as input the down-sampled nodes and associated features. In CPB (**a**), following [26, 12], the attention [33] mechanism is leveraged to incorporate more global contexts to the learned features. Following [12], we adopt a sequence of self-, cross- and self-attention modules, which interactively aggregates global contexts across nodes from the same and the other frame in a pair of point clouds. Below we briefly introduce the cross-attention module. Given $(\mathbf{F}'_{\mathbf{X}}, \mathbf{F}'_{\mathbf{Y}})$, akin to database retrieval, the former is linearly projected by a learnable matrix $\mathbf{W}_{\mathbf{Q}} \in \mathbb{R}^{b \times b}$ to *query* $\mathbf{Q}$ as $\mathbf{Q} = \mathbf{F}'_{\mathbf{X}} \mathbf{W}_{\mathbf{Q}}$, while the latter is similarly projected to *key* $\mathbf{K}$ and *value* $\mathbf{V}$ by learnable matrices $\mathbf{W}_{\mathbf{K}}$ and $\mathbf{W}_{\mathbf{V}}$, respectively. The attention matrix $\mathbf{A}$ is represented as $\mathbf{A} = \mathbf{Q}\mathbf{K}^T / \sqrt{b}$, whose rows are then normalized by *softmax*. The message $\mathbf{M}$ flows from $\mathbf{F}'_{\mathbf{Y}}$ to $\mathbf{F}'_{\mathbf{X}}$ is formulated as $\mathbf{M} = \mathbf{A} \cdot \mathbf{V}$, which represents the linear combination of *values* weighted by the attention matrix. In the cross-attention module, contexts are aggregated bidirectionally, from $\mathbf{F}'_{\mathbf{X}}$ to $\mathbf{F}'_{\mathbf{Y}}$ and from $\mathbf{F}'_{\mathbf{Y}}$ to $\mathbf{F}'_{\mathbf{X}}$. For computational efficiency, we replace the graph-based module [35] leveraged in [12] with the self-attention module in [26], which has the same architecture as the cross-attention module but takes the features from the same point cloud, e.g., $(\mathbf{F}'_{\mathbf{X}}, \mathbf{F}'_{\mathbf{X}})$, as input.

**Correspondence proposal.** As shown in CPB (**b**) of Fig. 1, we leverage strengthened features $\tilde{\mathbf{F}}'_{\mathbf{X}}$ and $\tilde{\mathbf{F}}'_{\mathbf{Y}}$ to calculate the similarity matrix. Down-sampled nodes whose vicinity areas share enough overlap are matched. However, there can be two cases where a node fails to match: 1) The major portion of its vicinity areas is occluded in the other frame. 2) Though most of its vicinity areas are visible in the other frame, there does not exist a node whose vicinity areas share sufficient overlap with its. Thus, for the similarity matrix, we expand it with a slack row and column with $m'$ and $n'$ slack entries, respectively [36]. So that nodes fail to match other nodes could match their corresponding slack entries, i.e., having maximum scores there. Similar to [26], we compute the similarity matrix using an inner product, which can be presented as:

$$\mathbf{S}' = \begin{bmatrix} \tilde{\mathbf{F}}'_{\mathbf{X}} \tilde{\mathbf{F}}'^{T}_{\mathbf{Y}} & \mathbf{z} \\ \mathbf{z}^T & z \end{bmatrix}, \qquad \mathbf{S}' \in \mathbb{R}^{(n'+1) \times (m'+1)}, \tag{1}$$

where all slack entries are set to the same learnable parameter $z$. On $\mathbf{S}'$ we run the Sinkhorn Algorithm [37, 38, 39], seeking an optimal solution for the optimal transport problem. After that, each entry $(i', j')$ in the obtained matrix represents the matching confidence between the node $i'$ and node $j'$ from $\mathbf{P}'_{\mathbf{X}}$ and $\mathbf{P}'_{\mathbf{Y}}$, respectively. To guarantee a higher recall, we adopt a threshold $\tau_c$ for likely correspondences whose confidence scores are above $\tau_c$. We define the obtained coarse node correspondence set as $\mathbf{C}' = \{(\mathbf{P}'_{\mathbf{X}}(i'), \mathbf{P}'_{\mathbf{Y}}(j'))\}$, with $|\mathbf{C}'| = c'$, where $|\cdot|$ denotes the set cardinality. Furthermore, we set the other threshold $\tau_m$ to guarantee that $c' \geq \tau_m$. When $c' < \tau_m$, we gradually decrease $\tau_c$ to extract more coarse node correspondences.

## 3.2 Point-level Refinement

**Node decoding.** On the finer scale, we aim at refining coarse correspondences from the preceding stage to point level. Those refined correspondences can then be used for point cloud registration. We first stack several KPConv [34] layers to recover the raw points, $\mathbf{P}_{\mathbf{X}}$ and $\mathbf{P}_{\mathbf{Y}}$, while jointly learning associated dense descriptors, $\mathbf{F}_{\mathbf{X}} \in \mathbb{R}^{n \times c}$ and $\mathbf{F}_{\mathbf{Y}} \in \mathbb{R}^{m \times c}$. We thereby assign to each point $p$ an associated feature $p \leftrightarrow f \in \mathbb{R}^c$, as illustrated in **3)** of Fig. 1. Then, as demonstrated in **4)** of Fig. 1, obtained dense descriptors, together with raw points and coarse correspondences are fed into the Correspondence Refinement Block (CRB), where coarse proposals are expanded to patches that are then refined to point correspondences.

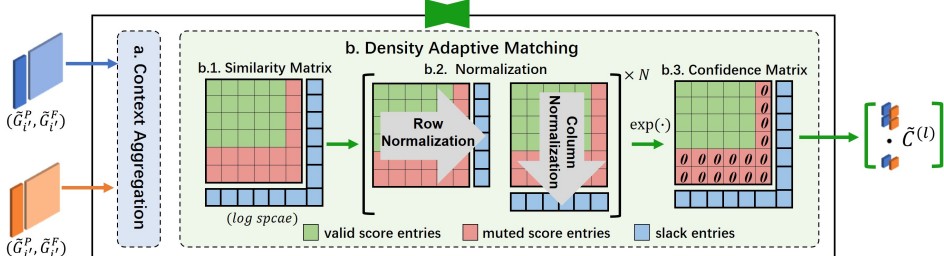

Figure 2: Illustration of our proposed density-adaptive matching module. The input is a pair of patches truncated by $k$. a) We use the context aggregation part from the Correspondence Proposal Block to condition on both patches and to strengthen features. b.1) The similarity matrix is computed. Slack entries are initialized with $0$ and muted entries corresponding to repeatedly sampled points are set to $-\infty$. b.2) N iterations of the Sinkhorn Algorithm are performed. We drop the slack row and column for row and column normalization, respectively. b.3) We obtain the confidence matrix, whose first $k$ rows and $k$ columns are row- and column-normalized, respectively. For correspondences, we pick the maximum confidence score in every row and column to guarantee a higher precision.

**Point-to-node grouping.** For refinement, we need to expand nodes in coarse correspondences to patches consisting of groups of points and associated descriptors. Accordingly, we use a point-to-node grouping strategy [40, 41, 22] to assign points to their nearest nodes in geometry space. If a point has multiple nearest nodes, a random one will be picked. We demonstrate this procedure in CRB (**a**) of Fig. 1. The advantages of point-to-node over k-nearest neighbor search or radius-based ball query are two-fold: 1) Every point will be assigned to exactly one node, while some points could be left out in other strategies. 2) It can automatically adapt to various scales [22]. After grouping, nodes with their associated points and descriptors form patches, upon which we can extract point correspondences. For a certain node $\mathbf{P}'_{\mathbf{X}}(i')$, its associated point set $\mathbf{G}^{\mathbf{P}}_{i'}$ and feature set $\mathbf{G}^{\mathbf{F}}_{i'}$ can be denoted as:

$$\begin{cases} \mathbf{G}^{\mathbf{P}}_{i'} = \{p \in \mathbf{P}_{\mathbf{X}} \big| \|p - \mathbf{P}'_{\mathbf{X}}(i')\| \le \|p - \mathbf{P}'_{\mathbf{X}}(j')\|, \forall j' \neq i'\}, \\ \mathbf{G}^{\mathbf{F}}_{i'} = \{f \in \mathbf{F}_{\mathbf{X}} \big| f \leftrightarrow p \text{ with } p \in \mathbf{G}^{\mathbf{P}}_{i'}\}, \end{cases} \quad (2)$$

where $\|\cdot\| = \|\cdot\|_2$ represents the Euclidean norm. In the point-to-node grouping, we expand the coarse node correspondence set $\mathbf{C}'$ to its corresponding patch correspondence set, both in geometry space $\mathbf{C_P} = \{(\mathbf{G}^{\mathbf{P}}_{i'}, \mathbf{G}^{\mathbf{P}}_{j'})\}$ and feature space $\mathbf{C_F} = \{(\mathbf{G}^{\mathbf{F}}_{i'}, \mathbf{G}^{\mathbf{F}}_{j'})\}$.

**Density-adaptive matching.** Extracting point correspondences from a pair of overlapped patches is in some way analogous to matching two smaller scale point clouds from a local perspective. Thus, directly leveraging the CPB in Fig. 1 with input $(\mathbf{G}^{\mathbf{P}}_{i'}, \mathbf{G}^{\mathbf{F}}_{i'})$ and $(\mathbf{G}^{\mathbf{P}}_{j'}, \mathbf{G}^{\mathbf{F}}_{j'})$ could theoretically tackle the problem. However, simply utilizing CPB to extract point correspondences would lead to a bias towards slack rows and columns, i.e., the model learns to predict more points as occluded. Reasons for this are two-fold: 1) For computational efficiency, similar to radius-based ball query, in a point-to-node grouping, we need to truncate the number of points to a unified number $k$ for every patch. If a patch contains less than $k$ points, like in [42], a fixed point or randomly sampled points will be repeated as a supplement. 2) On a coarse level, our model learns to propose corresponding nodes with overlapped vicinity areas. However, after expansion, proposed patches can be supplemented by some occluded points, which introduces biases in the training of refinement. To address the issue, we propose a density-adaptive matching module that refines coarse correspondences to point level by solving an optimal transport problem with awareness to point density. We denote the truncated patches as $\widetilde{\mathbf{G}}$ and demonstrate our proposed density-adaptive matching module in Fig. 2. Notably, both during and after normalization, the exponent projection of any muted entries is always equal to $0$, which eliminates the side effects caused by the repeated sampling of points. The final point correspondence set $\widetilde{\mathbf{C}}$ is represented as the union of all the obtained correspondence sets $\widetilde{\mathbf{C}}^{(l)}$. $\widetilde{\mathbf{C}}$ can be directly leveraged by RANSAC[21] for registration.

### 3.3 Loss Functions

Our total loss $\mathcal{L} = \mathcal{L}_c + \lambda\mathcal{L}_f$ is calculated as the weighted sum of the coarse-scale $\mathcal{L}_c$ and the fine-scale $\mathcal{L}_f$, where $\lambda$ is used to balance the terms. We detail the individual parts hereafter.

**Coarse scale.** On the coarse scale, we leverage a weighting scheme proportional to the overlap ratios over patches as coarse supervision. Given a pair of down-sampled nodes $\mathbf{P'_X}(i')$ and $\mathbf{P'_Y}(j')$, with their expanded patch representation in geometry space, $\mathbf{G}^{\mathbf{P}}_{i'}$ and $\mathbf{G}^{\mathbf{P}}_{j'}$, we can compute the ratio of points in $\mathbf{G}^{\mathbf{P}}_{i'}$ that are visible in point cloud $\mathbf{P_Y}$ as:

$$r(i') = \frac{|\{\mathbf{p} \in \mathbf{G}^{\mathbf{P}}_{i'}|\exists \mathbf{q} \in \mathbf{P_Y} \text{ s.t. } \|\overline{\mathbf{T}}^{\mathbf{X}}_{\mathbf{Y}}(\mathbf{p}) - \mathbf{q}\| < \tau_p\}|}{|\mathbf{G}^{\mathbf{P}}_{i'}|}, \tag{3}$$

where $\tau_p$ is the distance threshold. Similarly, we can calculate the ratio of points in $\mathbf{G}^{\mathbf{P}}_{i'}$ that have correspondences in $\mathbf{G}^{\mathbf{P}}_{j'}$ as:

$$r(i', j') = \frac{|\{\mathbf{p} \in \mathbf{G}^{\mathbf{P}}_{i'}|\exists \mathbf{q} \in \mathbf{G}^{\mathbf{P}}_{j'} \text{ s.t. } \|\overline{\mathbf{T}}^{\mathbf{X}}_{\mathbf{Y}}(\mathbf{p}) - \mathbf{q}\| < \tau_p\}|}{|\mathbf{G}^{\mathbf{P}}_{i'}|}. \tag{4}$$

Based on Eq. 3 and Eq. 4, we define the weight matrix $\mathbf{W}' \in \mathbb{R}^{(n'+1)\times(m'+1)}$ as:

$$\mathbf{W}'(i', j') = \begin{cases} min(r(i', j'), r(j', i')), & i' \leq n' \wedge j' \leq m', \\ 1 - r(i'), & i' \leq n' \wedge j' = m' + 1, \\ 1 - r(j'), & i' = n' + 1 \wedge j' \leq m', \\ 0, & \text{otherwise.} \end{cases} \tag{5}$$

Finally, we define the coarse scale loss as:

$$\mathcal{L}_c = \frac{-\sum_{i',j'} \mathbf{W}'(i', j') \log(\mathbf{S}'(i', j'))}{\sum_{i',j'} \mathbf{W}'(i', j')}. \tag{6}$$

**Finer scale.** On the finer point level, for the $l^{th}$ truncated patch correspondence $(\widetilde{\mathbf{G}}^{\mathbf{P}}_{i'}, \widetilde{\mathbf{G}}^{\mathbf{P}}_{j'})$ s.t. $(\mathbf{G}^{\mathbf{P}}_{i'}, \mathbf{G}^{\mathbf{P}}_{j'}) \in \mathbf{C_P}$, we define the binary matrix $\widetilde{\mathbf{B}}^{(l)} \in \mathbb{R}^{(k+1)\times(k+1)}$ as:

$$\widetilde{\mathbf{B}}^{(l)}(i, j) = \begin{cases} 1, & \|\overline{\mathbf{T}}^{\mathbf{X}}_{\mathbf{Y}}(\widetilde{\mathbf{G}}^{\mathbf{P}}_{i'}(i)) - \widetilde{\mathbf{G}}^{\mathbf{P}}_{j'}(j)\| < \tau_p, \\ 0, & \text{otherwise,} \end{cases} \qquad \forall i, \forall j \in [1, k], \tag{7}$$

and

$$\widetilde{\mathbf{B}}^{(l)}(i, k+1) = \max(0, 1 - \sum_{j=1}^{k} \widetilde{\mathbf{B}}^{(l)}(i, j)), \qquad \forall i \in [1, k],$$

$$\widetilde{\mathbf{B}}^{(l)}(k+1, j) = \max(0, 1 - \sum_{i=1}^{k} \widetilde{\mathbf{B}}^{(l)}(i, j)), \qquad \forall j \in [1, k]. \tag{8}$$

Additionally, we further set the rows and columns of $\widetilde{\mathbf{B}}^{(l)}$ which correspond to repeatedly sampled points to 0 to eliminate their side effects during training. $\widetilde{\mathbf{B}}^{(l)}(k+1, k+1)$ is also set to 0. Therefore, by defining the confidence matrix in **b.3** of Fig. 2 as $\widetilde{\mathbf{S}}^{(l)}$, the loss function on the finer scale reads as:

$$\mathcal{L}_f = \frac{-\sum_{l,i,j} \widetilde{\mathbf{B}}^{(l)}(i, j) \log(\widetilde{\mathbf{S}}^{(l)}(i, j))}{\sum_{l,i,j} \widetilde{\mathbf{B}}^{(l)}(i, j)}, \tag{9}$$

where we define $0 \cdot \log(0) = 0$.

## 4   Results

We evaluate our model on three challenging public benchmarks, including both indoor[1] and outdoor scenarios. Following [12], for indoor scenes, we evaluate our model on both 3DMatch [6], where point cloud pairs share > 30% overlap, and 3DLoMatch [12], where point cloud pairs have 10% ~30% overlap. In line with existing works [10, 12], we evaluate for outdoor scenes on odometryKITTI [20]. Please refer to Appendix for more details of implementation and datasets.

---

[1]As PREDATOR fixed a bug after our submission, we update their latest results. We also update ours according to the rebuttal.

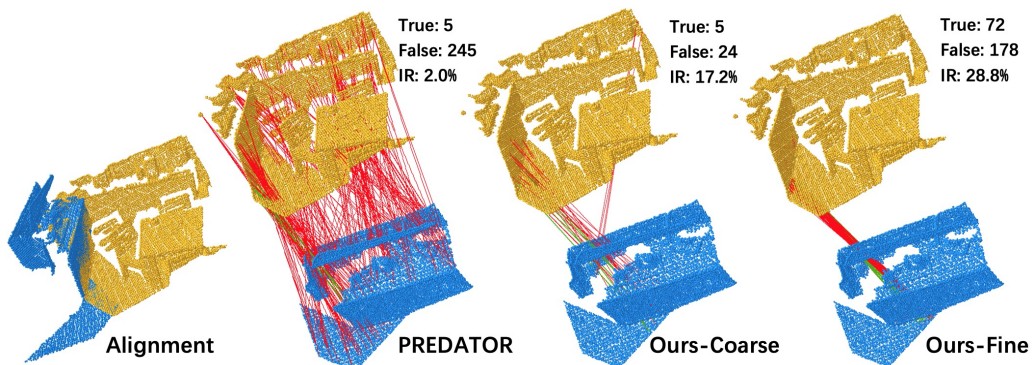

Figure 3: Qualitative results on *Inlier Ratio*. We compare our point correspondences (the last column) with our coarse correspondences (the third column) and correspondences from PREDATOR (the second column) on a hard case from 3DLoMatch. The first column provides the ground truth alignment, which shows that overlap is very limited. The significantly larger inlier ratio can be observed from the incorrect (red) and correct (green) correspondence connections.

## 4.1 3DMatch and 3DLoMatch

We compare our proposed CoFiNet to other state-of-the-art approaches including 3DSN [9], FCGF [31], D3Feat [10] and PREDATOR [12] in Tab. 1 [2] and Fig. 4. Comparisons to SpinImages [1], SHOT [2], FPFH [4] and 3DMatch [6] are also included in Fig. 4. For efficiency and qualitative results, please refer to Appendix.

Table 1: Results[2] on both 3DMatch and 3DLoMatch datasets under different numbers of samples. We also show the number of utilized parameters of all the approaches in the last column. Best performance is highlighted in bold while the second best is marked with an underline.

| # Samples | 3DMatch | | | | | 3DLoMatch | | | | | # Params ↓ |
|---|---|---|---|---|---|---|---|---|---|---|---|
| | 5000 | 2500 | 1000 | 500 | 250 | 5000 | 2500 | 1000 | 500 | 250 | |
| *Registration Recall(%)* ↑ | | | | | | | | | | | |
| 3DSN[9] | 78.4 | 76.2 | 71.4 | 67.6 | 50.8 | 33.0 | 29.0 | 23.3 | 17.0 | 11.0 | 10.2M |
| FCGF[31] | 85.1 | 84.7 | 83.3 | 81.6 | 71.4 | 40.1 | 41.7 | 38.2 | 35.4 | 26.8 | 8.76M |
| D3Feat[10] | 81.6 | 84.5 | 83.4 | 82.4 | 77.9 | 37.2 | 42.7 | 46.9 | 43.8 | 39.1 | 27.3M |
| PREDATOR[12] | 89.0 | **89.9** | **90.6** | **88.5** | 86.6 | 59.8 | 61.2 | 62.4 | 60.8 | 58.1 | 7.43M |
| CoFiNet(*ours*) | **89.3** | 88.9 | 88.4 | 87.4 | **87.0** | **67.5** | **66.2** | **64.2** | **63.1** | **61.0** | **5.48**M |
| *Feature Matching Recall(%)* ↑ | | | | | | | | | | | |
| 3DSN[9] | 95.0 | 94.3 | 92.9 | 90.1 | 82.9 | 63.6 | 61.7 | 53.6 | 45.2 | 34.2 | 10.2M |
| FCGF[31] | 97.4 | 97.3 | 97.0 | 96.7 | 96.6 | 76.6 | 75.4 | 74.2 | 71.7 | 67.3 | 8.76M |
| D3Feat[10] | 95.6 | 95.4 | 94.5 | 94.1 | 93.1 | 67.3 | 66.7 | 67.0 | 66.7 | 66.5 | 27.3M |
| PREDATOR[12] | 96.6 | 96.6 | 96.5 | 96.3 | 96.5 | 78.6 | 77.4 | 76.3 | 75.7 | 75.3 | 7.43M |
| CoFiNet(*ours*) | **98.1** | **98.3** | **98.1** | **98.2** | **98.3** | **83.1** | **83.5** | **83.3** | **83.1** | **82.6** | **5.48**M |
| *Inlier Ratio(%)* ↑ | | | | | | | | | | | |
| 3DSN[9] | 36.0 | 32.5 | 26.4 | 21.5 | 16.4 | 11.4 | 10.1 | 8.0 | 6.4 | 4.8 | 10.2M |
| FCGF[31] | 56.8 | 54.1 | 48.7 | 42.5 | 34.1 | 21.4 | 20.0 | 17.2 | 14.8 | 11.6 | 8.76M |
| D3Feat[10] | 39.0 | 38.8 | 40.4 | 41.5 | 41.8 | 13.2 | 13.1 | 14.0 | 14.6 | 15.0 | 27.3M |
| PREDATOR[12] | **58.0** | **58.4** | **57.1** | **54.1** | 49.3 | **26.7** | **28.1** | **28.3** | **27.5** | 25.8 | 7.43M |
| CoFiNet(*ours*) | 49.8 | 51.2 | 51.9 | 52.2 | **52.2** | 24.4 | 25.9 | 26.7 | 26.8 | **26.9** | **5.48**M |

**Metrics.** We adopt three typically-used metrics, namely *Registration Recall* (RR), *Feature Matching Recall* (FMR) and *Inlier Ratio* (IR), to show the superiority of CoFiNet over existing approaches. Specifically, 1) the *Registration Recall* is the fraction of point cloud pairs whose error of transformation estimated by RANSAC is smaller than a certain threshold, e.g., RMSE < 0.2m, compared to the ground truth. 2) The *Feature Matching Recall* indicates the percentage of point cloud pairs whose *Inlier Ratio* is larger than a certain threshold, e.g., $\tau_2 = 5\%$. 3) The *Inlier Ratio* is the fraction of correspondences whose residual error in geometry space is less than a threshold, e.g., $\tau_1 = 10cm$, under the ground truth transformation. Metric details are given in Appendix.

---

[2]As PREDATOR computes *Inlier Ratio* on a correspondence set different to the one used for registration, we give more results in 4.1 and Tab. 3 for a fair comparison.

**Correspondence sampling.**    We follow [10, 12] and report performance with different numbers of samples. However, as CoFiNet avoids keypoint detection and directly outputs point correspondences, we cannot strictly follow [10, 12] to sample different numbers of interest points. For a fair comparison, we instead sample correspondences in our experiments but keep the same number as them. Correspondences are sampled with probability proportional to a global confidence $c_{global} = c_{fine} \cdot c_{coarse}$. For a certain point correspondence refined from patch correspondence $(\widetilde{\mathbf{G}}^{\mathbf{P}}_{i'}, \widetilde{\mathbf{G}}^{\mathbf{P}}_{j'})$, we define $c_{fine}$ as its fine-level confidence score and $c_{coarse}$ as $\mathbf{S}'(i', j')$.

***Inlier Ratio.***[2]    As the main contribution of CoFiNet is that we adopt the coarse-to-fine mechanism to avoid keypoint detection, while existing methods struggle to sample repeatable keypoints for matching, we first check the *Inlier Ratio* of CoFiNet, which is directly related to the quality of extracted correspondences. We show quantitative results in Tab. 1 and qualitative results in Fig. 3. As shown in Tab. 1, on *Inlier Ratio*, CoFiNet outperforms all the previous methods except PREDATOR [12] on 3DLoMatch and only performs worse than PREDATOR [12] and FCGF[31] on 3DMatch. Notably, when the sample number is 250, we perform the best on both datasets, since detection-based methods face a more severe repeatability problem in this case. By contrast, as our method leverages a coarse-to-fine mechanism and thus avoids keypoint detection, it is more robust to the aforementioned case. Furthermore, the fact that sampling fewer correspondences leads to a higher *Inlier Ratio* indicates that our learned scores are well-calibrated, i.e., higher confidence scores indicate more reliable correspondences.

Table 2: Registration results without RANSAC [21]. Relative poses are directly solved based on extracted correspondences by singular value decomposition (SVD). Best performance is highlighted in bold while the second best is marked with an underline.

| # Samples | 3DMatch | | | | | 3DLoMatch | | | | |
|---|---|---|---|---|---|---|---|---|---|---|
| | 5000 | 2500 | 1000 | 500 | 250 | 5000 | 2500 | 1000 | 500 | 250 |
| *Registration Recall w/o RANSAC (%)* ↑ | | | | | | | | | | |
| FCGF[31] | 28.5 | 27.9 | 25.7 | 23.2 | 21.2 | 2.3 | 1.7 | 1.3 | 1.1 | 1.1 |
| D3Feat[10] | 24.3 | 24.0 | 23.0 | 22.4 | 19.1 | 1.1 | 1.4 | 1.1 | 1.0 | 1.0 |
| PREDATOR[12] | 48.7 | 51.8 | 54.3 | 53.5 | 53.0 | 6.1 | 8.1 | 10.1 | 11.4 | 11.3 |
| CoFiNet(*ours*) | **63.2** | **63.4** | **63.8** | **64.9** | **64.6** | **19.0** | **20.4** | **21.0** | **20.9** | **21.6** |

**Reliability of our correspondences.**    Though *Inlier Ratio* is an important metric of correspondence quality, it is naturally affected by the distance threshold $\tau_1$. To better illustrate the reliability of correspondences extracted by CoFiNet and show our superiority over existing methods, we conduct another experiment and show related results in Tab. 2. In this experiment, we directly solve the relative poses using singular value decomposition (SVD) based on extracted correspondences, without the assistance of the robust estimator RANSAC [21]. As we can see, for FCGF [31] and D3Feat [10], though they can work on 3DMatch, they fail on 3DLoMatch, where point cloud share less overlap and thus reliable correspondences are harder to obtain. Compared with PREDATOR [12], on both 3DMatch and 3DLoMatch, our proposed CoFiNet performs much better, which indicates that we propose more reliable correspondences on both datasets.

***Feature Matching Recall*** **and** ***Registration Recall.***    On *Feature Matching Recall*, CoFiNet significantly outperforms all the other methods on both 3DMatch and 3DLoMatch. Especially on 3DLoMatch, which is more challenging due to the low-overlap scenarios, our proposed method surpasses others with a large margin of more than 4%. It indicates that CoFiNet is more robust to different scenes, i.e., we find at least 5% inlier correspondences for more test cases. Additionally, we also follow [10, 12] to show the *Feature Matching Recall* in relation to $\tau_2$ and $\tau_1$ on 3DMatch in

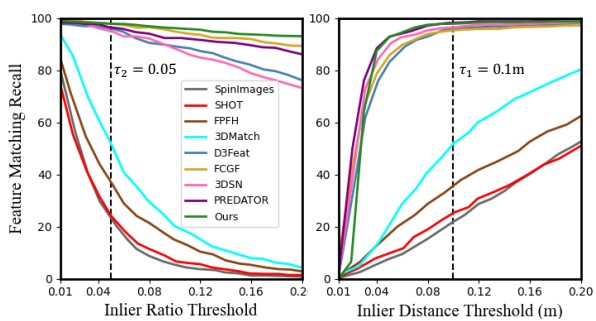

Figure 4: *Feature Matching Recall* in relation to: 1) *Inlier Ratio Threshold* ($\tau_2$) and 2) *Inlier Distance Threshold* ($\tau_1$) on 3DMatch.

Fig. 4, which further shows our superiority over other methods. When referring to the most important metric *Registration Recall* which better reflects the final performance on point cloud registration, though we perform slightly worse than PREDATOR [12], we significantly outperform others on 3DMatch. When evaluated on 3DLoMatch, our proposed approach significantly surpasses all the others, which shows the advantages of our method in scenarios with less overlap. Moreover, we also compare the number of parameters used in different methods in the last column of Tab.1, which shows that CoFiNet uses the least parameters while achieving the best performance.

Table 3: *Inlier Ratio* and *Registration Recall* on the same correspondence set. For CoFiNet, coarse correspondences are extracted based on thresholds and *non-mutual* selection is used on the finer scale. Best performance is highlighted in bold while the second best is marked with an underline.

| # Samples | 3DMatch | | | | | 3DLoMatch | | | | |
|---|---|---|---|---|---|---|---|---|---|---|
| | 5000 | 2500 | 1000 | 500 | 250 | 5000 | 2500 | 1000 | 500 | 250 |
| | *Registration Recall(%)* ↑ | | | | | | | | | |
| PREDATOR[12](*mutual*) | 86.6 | 86.4 | 85.3 | 85.6 | 84.3 | 61.8 | 61.8 | 61.6 | 58.4 | 56.2 |
| PREDATOR[12](*non-mutual*) | 89.0 | **89.9** | **90.6** | **88.5** | 86.6 | 59.8 | 61.2 | 62.4 | 60.8 | 58.1 |
| CoFiNet(*ours*) | **89.3** | 88.9 | 88.4 | 87.4 | **87.0** | **67.5** | **66.2** | **64.2** | **63.1** | **61.0** |
| | *Inlier Ratio(%)* ↑ | | | | | | | | | |
| PREDATOR[12](*mutual*) | **58.0** | **58.4** | **57.1** | **54.1** | 49.3 | **26.7** | **28.1** | **28.3** | **27.5** | 25.8 |
| PREDATOR[12](*non-mutual*) | 46.6 | 48.3 | 47.2 | 44.1 | 38.8 | 19.3 | 21.6 | 22.1 | 21.3 | 19.7 |
| CoFiNet(*ours*) | 49.8 | 51.2 | 51.9 | 52.2 | **52.2** | 24.4 | 25.9 | 26.7 | 26.8 | **26.9** |

***Inlier Ratio* and *Registration Recall* on the same correspondence set.** In Tab. 1, PREDATOR [12] reports *Inlier Ratio* on a correspondence set that is different to the one used for registration, while CoFiNet uses the same. PREDATOR uses correspondences extracted by *mutual* selection to report *Inlier Ratio*, while computing *Registration Recall* on a correspondence set obtained by *non-mutual* selection. As we target at registration, we consider it meaningless to evaluate on a correspondence set that is not used for pose estimation. Thus, to make a fair comparison, we compare CoFiNet with both PREDATOR(*mutual*) and PREDATOR(*non-mutual*) in Tab. 3. In *mutual* selection, two points **x** and **y** are considered as a correspondence when **x** match to **y** **and** **y** match to **x**, while in *non-mutual* selection, the correspondence is extracted when **x** match to **y** **or** **y** match to **x**. In Tab. 3, compared to *non-mutual*, *mutual* selection rejects some outliers, and thus increases *Inlier Ratio* of PREDATOR. However, as it meanwhile filters out some inlier correspondences, when combined with RANSAC [21], *Registration Recall* usually drops. Since our task is registration, PREDATOR(*non-mutual*) with higher *Registration Recall* is preferred over itself with *mutual* selection. In this case, CoFiNet achieves higher *Inlier Ratio* than PREDATOR on both datasets.

**Influence of the number of coarse correspondences.** As illustrated in Tab. 4, on both 3DMatch and 3DLo-Match, when sampled only with $\tau_c$, a higher threshold results in fewer coarse correspondences and meanwhile a higher *Inlier Ratio*, which indicates the learned

Table 4: Ablation study of the number of coarse correspondences, tested with # Samples=2500. # Coarse indicates the average number of sampled coarse correspondences. Best performance is highlighted in bold.

| $\tau_c$ | $\tau_m$ | 3DMatch | | | 3DLoMatch | | |
|---|---|---|---|---|---|---|---|
| | | IR (%)↑ | RR (%) ↑ | # Coarse | IR (%)↑ | RR (%)↑ | # Coarse |
| 0.05 | - | 49.4 | 87.4 | 575 | 27.3 | 62.8 | 260 |
| 0.10 | - | 51.1 | 88.1 | 335 | 29.8 | 62.3 | 128 |
| 0.15 | - | **55.5** | 85.5 | 222 | **32.7** | 58.9 | 74 |
| 0.20 | 200 | 51.2 | **88.9** | 230 | 25.9 | **66.2** | 203 |

confidence scores are well-calibrated on the coarse level. However, *Registration Recall* drops at the same time, as the number of correspondences for refinement is decreased, and thus fewer point correspondences are leveraged in RANSAC for pose estimation. The last row is the strategy used in our paper. Except for $\tau_c$, we also use $\tau_m$ to guarantee that CoFiNet samples at least $\tau_m$ coarse correspondences on each point cloud pair, as described before. This strategy slightly sacrifices *Inlier Ratio* but brings significant improvements on *Registration Recall*.

**Importance of individual modules.** As shown in Tab. 5, in the first experiment, we directly use the coarse correspondence set $\mathbf{C}'$ for point cloud registration. Unsurprisingly, it performs worse on

all the metrics, indicating that CoFiNet benefits from refinement. Then, we ablate the weighting scheme which is proportional to overlap ratios and guides the coarse matching of down-sampled nodes. We replace it with a binary mask similar to the one used on the finer level. Results show that it leads to a worse performance, which proves that coarse matching of nodes benefits from our designed weighting scheme. Finally, we do the last ablation study on the density-adaptive matching module. Results indicate that on both 3DMatch and 3DLoMatch, with the density-adaptive matching module, CoFiNet better adapts to the irregular nature of point clouds.

Table 5: Ablation study of individual modules, tested with # Samples=2500. Best performance is highlighted in bold.

|  | 3DMatch | | | 3DLoMatch | | |
|---|---|---|---|---|---|---|
|  | RR (%)↑ | FMR (%)↑ | IR (%)↑ | RR (%)↑ | FMR (%)↑ | IR (%)↑ |
| Full CoFiNet | **88.9** | **98.3** | **51.2** | **66.2** | **83.5** | **25.9** |
| w/o refinement | 79.6 | 96.5 | 44.3 | 41.2 | 81.4 | 21.3 |
| w/o weighting | 87.4 | 97.3 | 50.0 | 61.5 | 80.5 | 23.5 |
| w/o density-adaptive | 88.3 | 97.9 | 49.3 | 65.1 | 82.7 | 24.7 |

## 4.2 KITTI

**Metrics.** We follow [12] and use 3 metrics, namely, the *Relative Rotation Error* (RRE), which is the geodesic distance between estimated and ground truth rotation matrices, the *Relative Translation Error* (RTE), which is the Euclidean distance between the estimated and ground truth translation, and the *Registration Recall* (RR) mentioned before. More details are provided in Appendix.

**Comparisons to existing approaches.** On KITTI, we compare CoFiNet to 3DFeat-net [43], FCGF [31], D3Feat [10] and PREDATOR [12]. Quantitative results can be found in Tab. 6, while qualitative results are given in Appendix. On RTE and RRE, we stay in the middle, but for RR, together with [10, 12], we perform the best. Notably, we achieve such performance by us-

Table 6: Quantitative comparisons on KITTI. Best performance is highlighted in bold.

| Method | RTE(cm)↓ | RRE(°)↓ | RR(%)↑ | Params↓ |
|---|---|---|---|---|
| 3DFeat-Net [43] | 25.9 | 0.57 | 96.0 | **0.32M** |
| FCGF [31] | 9.5 | 0.30 | 96.6 | 8.76M |
| D3Feat [10] | 7.2 | 0.30 | **99.8** | 27.3M |
| PREDATOR [12] | **6.8** | **0.27** | **99.8** | 22.8M |
| CoFiNet(ours) | 8.5 | 0.41 | **99.8** | 5.48M |

ing only 5.48M parameters and training for 20 epochs compared to the best performing model [12], which uses over 20M parameters and is trained for 150 epochs. This experiment indicates that our model can deal with outdoor scenarios.

## 5 Conclusion

In this paper, we present a deep neural network that leverages a coarse-to-fine strategy to extract correspondences from unordered and irregularly sampled point clouds for registration. Our proposed model is capable of directly consuming unordered point sets and proposing reliable correspondences without the assistance of keypoints. To tackle the irregularity of point clouds, on a coarse scale, we first propose a weighting scheme proportional to local overlap ratios. It guides the model to match nodes that have overlapped vicinity areas, which significantly shrinks the search space of the following refinement. On a finer level, we then adopt a density-adaptive matching module, which eliminates the side effects from repeated sampling and enables our model to deal with density varying points. Extensive experiments on both indoor and outdoor benchmarks validate the effectiveness of our proposed model. We stay on par with the state-of-the-art approaches on 3DMatch and KITTI, while surpassing them on 3DLoMatch using a model with significantly fewer parameters. Limitations and broader impact are discussed in Appendix.

## Acknowledgments

We appreciate the valuable discussion with Zheng Qin and Hao Wu. We would like to thank Dr. Kai Wang for paper revision. Hao Yu is supported by China Scholarship Council (CSC).

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
