# A Appendix

In this supplementary material, we first provide detailed network architectures in Sec. A.1. Then details of metrics utilized in our experiments are demonstrated in Sec. A.2. We give more implementation details in Sec. A.3. We further introduce our utilized datasets in Sec. A.4 and evaluate the inference time of CoFiNet in Sec. A.5. Limitations and broader impact are then discussed in Sec. A.6 and Sec. A.7, respectively. Finally, qualitative results of registration are provided in Sec. A.8.

## A.1 Network Architectures

CoFiNet mainly leverages an encoder-decoder architecture based on KPConv [1] operations, where we also add two attention-based networks [2] for context aggregation. Details of our network architecture are demonstrated in Fig. 1. Compared to [3], though we add additional local attention layers, our coarse-to-fine design enables us to use a lightweight encoder, which leads to the reduction of around 2M and over 20M parameters on 3DMatch/3DLoMatch and KITTI, respectively. Since we use the voxel size and convolution radius same to PREDATOR [3] for our KPConv backbone, each time of point down-sampling in CoFiNet results in nodes identical to that in [3].

## A.2 Evaluation Metrics

**Inlier Ratio** *Inlier Ratio* (IR) measures the fraction of point correspondences $(x_i, y_j) \in \widetilde{\mathbf{C}}$ s.t. the Euclidean Norm of residual $\|\overline{\mathbf{T}}_{\mathbf{Y}}^{\mathbf{X}}(x_i) - y_j\|$ is within a certain threshold $\tau_1$=10cm, where $\overline{\mathbf{T}}_{\mathbf{Y}}^{\mathbf{X}}$ indicates the ground truth transformation between $\mathbf{X}$ and $\mathbf{Y}$. Given the estimated correspondence set $\widetilde{\mathbf{C}}$, *Inlier Ratio* of a single point cloud pair $(\mathbf{X}, \mathbf{Y})$ can be calculated by:

$$\text{IR}(\mathbf{X}, \mathbf{Y}) = \frac{1}{|\widetilde{\mathbf{C}}|} \sum_{(x_i, y_j) \in \widetilde{\mathbf{C}}} \mathbb{1}(\|\overline{\mathbf{T}}_{\mathbf{Y}}^{\mathbf{X}}(x_i) - y_j\| < \tau_1), \tag{1}$$

where $\mathbb{1}(\cdot)$ represents the indicator function and $\|\cdot\| = \|\cdot\|_2$ denotes the Euclidean Norm.

**Feature Matching Recall** *Feature Matching Recall* (FMR) measures the fraction of point cloud pairs whose *Inlier Ratio* is larger than a certain threshold $\tau_2 = 5\%$. It is first utilized in [4] and it indicates the likelihood that the optimal transformation between two point clouds can be recovered by a robust pose estimator, e.g., RANSAC [5], based on the predicted correspondence set $\widetilde{\mathbf{C}}$. Given a dataset $\mathcal{D}$ with $|\mathcal{D}|$ point cloud pairs, *Feature Matching Recall* can be represented as:

$$\text{FMR}(\mathcal{D}) = \frac{1}{|\mathcal{D}|} \sum_{(\mathbf{X}, \mathbf{Y}) \in \mathcal{D}} \mathbb{1}(\text{IR}(\mathbf{X}, \mathbf{Y}) > \tau_2). \tag{2}$$

**Registration Recall** Different from the aforementioned metrics which measure the quality of extracted correspondences, *Registration Recall* (RR) directly measures the performance on our target task of point cloud registration. It measures the fraction of point cloud pairs whose Root Mean Square Error (RMSE) is within a certain threshold $\tau_3 = 0.2$m. Give a dataset $\mathcal{D}$ with $|\mathcal{D}|$ point cloud pairs, *Registration Recall* is defined as:

$$\text{RR}(\mathcal{D}) = \frac{1}{|\mathcal{D}|} \sum_{(\mathbf{X}, \mathbf{Y}) \in \mathcal{D}} \mathbb{1}(\text{RMSE}(\mathbf{X}, \mathbf{Y}) < \tau_3), \tag{3}$$

where for each $(\mathbf{X}, \mathbf{Y}) \in \mathcal{D}$, RMSE of the ground truth correspondence set $\overline{\mathbf{C}}$ after applying the estimated transformation $\mathbf{T}_{\mathbf{Y}}^{\mathbf{X}}$ reads as:

$$\text{RMSE}(\mathbf{X}, \mathbf{Y}) = \sqrt{\frac{1}{|\overline{\mathbf{C}}|} \sum_{(x_i, y_j) \in \overline{\mathbf{C}}} \|\mathbf{T}_{\mathbf{Y}}^{\mathbf{X}}(x_i) - y_j\|^2}. \tag{4}$$

Additionally, we follow the original evaluation protocol in 3DMatch [6], which excludes immediately adjacent point clouds with very high overlap ratios.

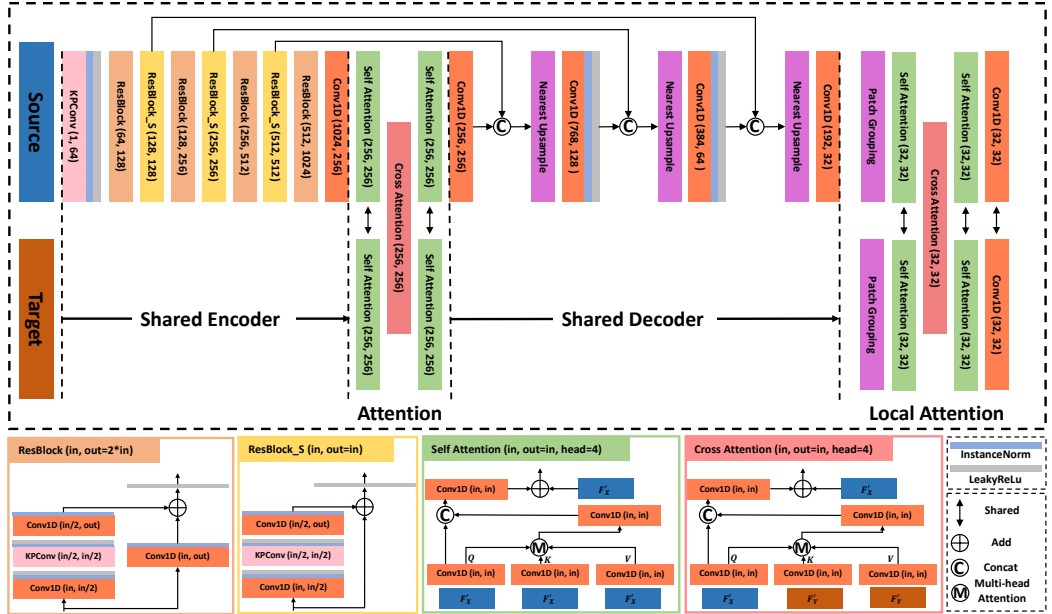

Figure 1: The detailed architecture of our proposed CoFiNet. In self- and cross-attention modules, we use four heads for the multi-head attention part. For instance, in self-attention modules (bottom centre), for *query* $\mathbf{Q} \in \mathbb{R}^{n' \times b}$, *key* $\mathbf{K} \in \mathbb{R}^{n' \times b}$ and *value* $\mathbf{V} \in \mathbb{R}^{n' \times b}$, we first reshape each of them into shape $(n', 4, \frac{b}{4})$ and then compute messages separately, which leads to messages in shape $(n', 4, \frac{b}{4})$. Finally we concatenate all the computed messages and obtain the message $\mathbf{M} \in \mathbb{R}^{n' \times b}$. The Patch Grouping layer indicates the Grouping module in the Correspondence Refinement Block. Self- and cross-attention modules (lower right) represent the case in the Attention part (centre), while in the Local Attention part (upper right), $\mathbf{F}'_{\mathbf{X}}$ and $\mathbf{F}'_{\mathbf{Y}}$ are replaced with $\widetilde{\mathbf{G}}^{\mathbf{F}}_{i'}$ and $\widetilde{\mathbf{G}}^{\mathbf{F}}_{j'}$, respectively.

**Relative Translation and Rotation Errors**   Given the estimated transformation $\mathbf{T}^{\mathbf{X}}_{\mathbf{Y}} \in SE(3)$ with a translation vector $\mathbf{t} \in \mathbb{R}^3$ and a rotation matrix $\mathbf{R} \in SO(3)$. Its Relative Translation Error (RTE) and Relative Rotation Error (RRE) from the ground truth pose $\overline{\mathbf{T}}^{\mathbf{X}}_{\mathbf{Y}}$ are computed as:

$$\text{RTE} = \|\mathbf{t} - \overline{\mathbf{t}}\| \qquad \text{and} \qquad \text{RRE} = \arccos(\frac{\text{trace}(\mathbf{R}^\top \overline{\mathbf{R}}) - 1}{2}), \tag{5}$$

where $\overline{\mathbf{t}}$ and $\overline{\mathbf{R}}$ are the the ground truth translation and rotation in $\overline{\mathbf{T}}^{\mathbf{X}}_{\mathbf{Y}}$, respectively.

### A.3   Implementation Details

CoFiNet is implemented in PyTorch [7] and can be trained end-to-end on a single RTX 2080Ti GPU. We train 20 epochs on 3DMatch/3DLoMatch and KITTI, with $\lambda = 1$, both using Adam optimizer with an initial learning rate of 3e-4, which is exponentially decayed by 0.05 after each epoch. We adopt similar encoder and decoder architectures as [3], but with significantly fewer parameters. We use a batch size of 1 in all experiments. For training the attention-based network on a finer scale, we sample 128 coarse correspondences, with truncated patch size $k = 64$ on 3DMatch (3DLoMatch). On KITTI, the numbers are 128 and 32, respectively. Moreover, due to the severely varying point density on KITTI, we only sample node correspondences with overlap ratios $> 20\%$ for training. At test time, all the extracted coarse correspondences are fed into the finer stage for refinement, with the same $k$ as in training. We use our proposed point correspondences and RANSAC [5] for registration.

### A.4   Data

**3DMatch and 3DLoMatch**   3DMatch [6] collects 62 scenes from SUN3D [8], 7-Scenes [9], RGB-D Scenes v.2 [10], Analysis-by-Synthesis [11], BundleFusion [12] and Halbel et al. [13], where 46 scenes are used for training, 8 scenes for validation and 8 scenes for testing. We utilize the training

data in [3] for training and also follow its evaluation protocols for testing. In training, input point cloud frames are generated by fusing 50 consecutive depth frames using TSDF volumetric fusion [14]. Different from the original 3DMatch [6] that only consists of point cloud pairs with >30% overlaps, in [3], point cloud pairs with overlaps between 10% and 30% are also included. Two benchmarks are leveraged for testing, namely, 3DMatch that consists of point cloud pairs with >30% overlaps, and 3DLoMatch which only includes point cloud pairs whose overlaps are between 10% and 30%. We also follow [3] to use voxel-grid down-sampling for preprocessing, where a random point will be picked when multiple points fall into the same voxel grid.

**OdometryKITTI** KITTI [15] is published under the NonCommercial-ShareAlike 3.0 License. It consists of 11 sequences scanned by a Velodyne HDL-64 3D laser scanner in driving scenarios. We follow [16] to pick point cloud pairs with at least 10m intervals from the raw data, which leads to 1,358 training pairs, 180 validation pairs, and 555 testing pairs. Moreover, as the ground truth poses provided by GPS are noisy, we follow [16] to use ICP to further refine them.

Table 1: Model runtime comparisons for a single inference. Time is averaged over the whole 3DMatch [6] testing set, which consists of 1,623 point cloud pairs. As our target task is registration and neural networks only provide intermediate results which are later consumed by RANSAC [5] for pose estimation, we also include the time of writing related results to hard disks.

| | CPU | GPU | Time(s)↓ | Improvement(%)↑ |
|---|---|---|---|---|
| PREDATOR [3] | i7-9700KF @ 3.60GHZ × 8 | GeForce RTX 3070 | 0.72 | - |
| CoFiNet(*ours*) | i7-9700KF @ 3.60GHZ × 8 | GeForce RTX 3070 | **0.25** | **65.3** |

## A.5 Timings

We further evaluate the inference time of CoFiNet and compare it to that of PREDATOR [3] which obtains the highest inference rate among all the state-of-the-art methods. Related results in Tab. 1 indicate the superiority of CoFiNet over PREDATOR in terms of computational efficiency. Notably, CoFiNet directly proposes point correspondences, while PREDATOR only outputs dense descriptors, and correspondences are extracted during RANSAC [5]. We further compare CoFiNet to PREDATOR in regard to RANSAC runtime, related results are illustrated in Tab. 2. Benefiting from our design, we reduce the RANSAC runtime significantly, especially when more correspondences are leveraged for pose estimation.

Table 2: RANSAC [5] runtime comparisons for a single inference. Time is averaged over the whole 3DMatch [6] testing set, which consists of 1,623 point cloud pairs. Settings are the same with Tab. 1

| # Samples | 5000 | 2500 | 1000 | 500 | 250 |
|---|---|---|---|---|---|
| PREDATOR [3] | 2.86s | 1.25s | 0.45s | 0.22s | 0.11s |
| CoFiNet(*ours*) | **0.18s** | **0.11s** | **0.07s** | **0.05s** | **0.05s** |

## A.6 Limitations

The limitations of our proposed CoFiNet are three-fold. 1) There is no explicit design for rejecting outliers from a coarse scale. False coarse correspondences can be expanded to false point correspondences which could result in lower *Inlier Ratio* on a finer level. As shown in column (c) and column (d) of the first row in Fig. 2, after refinement, the *Inlier Ratio* drops. 2) CoFiNet is challenged by those non-distinctive regions. As illustrated in column (d) of the first row in Fig. 2, mismatched points are located on the surface of the table, which is a flat area with little variability. 3) Point correspondences expanded from coarse correspondences are not sparse enough, which might introduce side effects to RANSAC[5] based point cloud registration. As demonstrated in column (d) of the second row in Fig. 2, in comparison to PREDATOR [3], our method produces a much better *Inlier Ratio* but extracts less sparser correspondences.

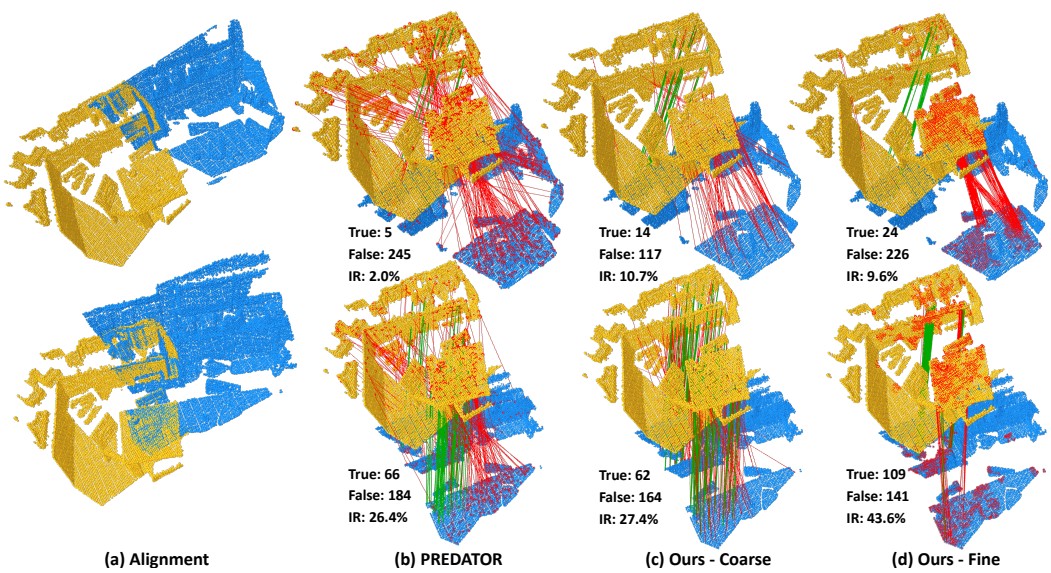

| (a) Alignment | (b) PREDATOR | (c) Ours - Coarse | (d) Ours - Fine |

Figure 2: Visualization of correspondences. Examples are from 3DLoMatch [3] and we compare our method to PREDATOR [3]. In column (b) and column (d), we only visualize 250 correspondences for better visibility but mark all the incorrectly matched points as red in both source and target point clouds. Correct correspondences are drawn in green.

## A.7 Broader Impact

We present a novel deep neural network that leverages the coarse-to-fine mechanism to extract correspondences from point clouds, which can be utilized for registration. It makes a first attempt towards the detection-free matching between a pair of unordered, irregular point sets. Our work can contribute to a wide range of applications, such as scene reconstruction, autonomous driving, simultaneous localization and mapping (SLAM), or any other where point cloud registration plays a role. For instance, the reconstruction of indoor scenes from unlabeled RGB-D images could benefit from our method, as it is capable of extracting reliable correspondences that can be leveraged to recover the rigid transformation between different frames precisely. Also, in autonomous driving scenarios, our methods can help agents better sense their surroundings. As our method aims at tackling a fundamental problem in computer vision, we do not anticipate a direct negative outcome. Potential negative outcomes might occur in real applications where our method is involved.

## A.8 Qualitative Results of Registration

Visualization of example registration from different datasets can be found in Fig. 3. Relative poses are estimated by RANSAC [5] that takes correspondences extracted by CoFiNet as input.

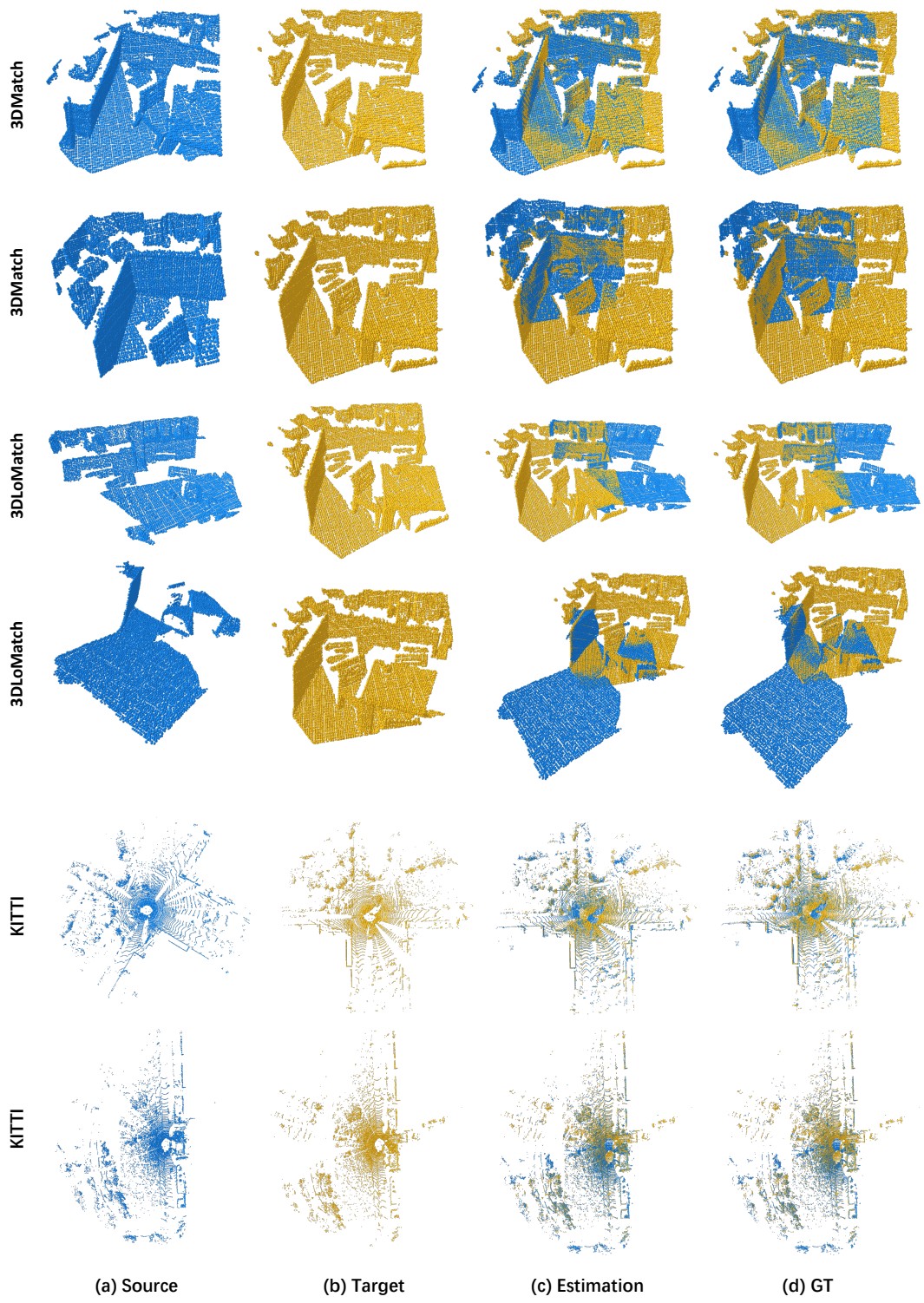

Figure 3: Qualitative registration results. We show two examples for each dataset. Column (a) and column (b) demonstrate the input point cloud pairs. Column (c) shows the estimated registration while column (d) provides the ground truth alignment.