# OpenReview forum: "CoFiNet: Reliable Coarse-to-fine Correspondences for Robust PointCloud Registration"
_NeurIPS.cc/2021/Conference — NeurIPS 2021 Poster_

### Official Review · Reviewer_5BJR · 2021-07-05

**Rating:** 6
**Confidence:** 4

**Summary:**

The authors present a coarse-to-fine approach to extract correspondences that can be used for a RANSAC-based point cloud registration.
They demonstrate on standardized datasets that they achieve a higher or similar inlier ratio, registration recall and feature matching recall than current state of the art with fewer parameters.

**Limitations And Societal Impact:**

Yes, as much as can be expected for the topic of this paper.

**Main Review:**

Interesting approach to employ a coarse-to-fine approach for refining matching points which improves on the two-stage RANSAC-based methods.

The paper is well-written though parts could be made clearer.


Some questions:

I'm a bit unclear on the changes you've made in the CPB block. Could you expand on how your correspondence proposal block compares to the one used in [24], is there a difference between their dustbins and your slack variables? And have you compared how the performance depends on the number of self- + cross-attention stacks you use (currently you use self- + cross- + self-attention)?

You've examined the use of a uniform sampling procedure as well as one based on the predicted scores but did you also try a voxel-based sampling procedure or similar to maintain the larger distribution of points that, as you discuss in the supplementary material, is necessary for a reliable registration?
Could this be the reason why the RR metric is similar with both the 'prob' and 'rand' sampling strategies while the IR is clearly in favor of the 'prob' approach?

The results on KITTI seems good, is the RTE and RRE calculated for the correctly registered point cloud or for all?

One comment is that the W and the B matrices used in the loss calculations probably should be defined in the paper not only verbally but similarly to what is done in the supplementary. It could also be clearer if the patch-to-patch correspondences are used or the point-to-point correspondences inside of two patches are compared since  $\tilde{S}^{(l)}$ is not defined.

Could you clarify the exact nature of the RANSAC procedure, is the GT transformation involved in selecting the correct transformation or is it only based on the number of inliers as would be done in a real-world scenario?


On the whole, it's a good paper that takes the concept of coarse-to-fine refinement from the 2D domain to the 3D case which results in a higher inlier ratio.

Update after rebuttal:

The authors addressed most issues from the reviews well. My already positive assessment remain unchanged.

**Time Spent Reviewing:**

7

---

> ### Author Response · Authors · 2021-08-10
> **Response to Reviewer 5BJR**
>
> We first would like to thank the reviewer for giving us valuable comments.
>
> 1. **Q**: I'm a bit unclear on the changes you've made in the CPB block. Could you expand on how your correspondence proposal block compares to the one used in [24], is there a difference between their dustbins and your slack variables? And have you compared how the performance depends on the number of self- + cross-attention stacks you use (currently you use self- + cross- + self-attention)?
>
>     **A**: Our CPB is similar to the one used in [24]. Dustbins and slack variables are the same. The main difference is that we use overlap ratios between patches as supervision and propose a weighting scheme guiding the training on the coarse level. We conclude this weighting scheme as one of our main contributions. Related ablation study can be found in Tab.2 (w/o weighting).
>
>     For the number of attention modules, in our paper, we use 3 (self- + cross- + self-attention). In the following table, we show two more cases, where the first one is w/o attention and the second one is with another self- + cross-attention (totally 5). In general, compared to the case w/o attention, using 3 attention modules brings improvements especially on 3DLoMatch. However, using 5 attention modules doesn’t help too much and it introduces 1M more parameters. We use the same threshold on the coarse level for all the experiments.
>
>     **Registration Recall**:
>
>     | | | |3DMatch | | | |3DLoMatch | | | |
>     |:-:|:-:|:-:|:-:|:-:|:-:|:-:|:-:|:-:|:-:|:-:|
>     |#Sampled Points|5000| 2500|1000|500|250|5000| 2500|1000|500|250|
>     |CoFiNet w/o attention|87.8%|87.9%|**87.9%**|**87.0%**|86.0%|61.8%|61.4%|60.3%|59.5%|58.5%|
>     |CoFiNet w/ 3 attention|87.4%|88.1%|86.6%|86.4%|85.9%|**63.6%**|62.3%|**61.8%**|60.3%|**60.2%**|
>     |CoFiNet w/ 5 attention|**89.6%**|**89.0%**|87.8%|**87.0%**|**87.6%**|63.4%|**63.1%**|61.0%|**60.6%**|59.8%|
>
>     **Inlier Ratio**:
>
>     | | | |3DMatch | | | |3DLoMatch | | | |
>     |:-:|:-:|:-:|:-:|:-:|:-:|:-:|:-:|:-:|:-:|:-:|
>     |#Sampled Points|5000| 2500|1000|500|250|5000| 2500|1000|500|250|
>     |CoFiNet w/o attention|47.3%|49.2%|50.1%|50.4%|50.5%|25.3%|26.5%|27.5%|27.9%|28.0%|
>     |CoFiNet w/ 3 attention|**50.4%**|**51.1%**|**51.6%**|**51.7%**|**52.0%**|**28.9%**|**29.8%**|**30.4%**|**30.6%**|**30.8%**|
>     |CoFiNet w/ 5 attention|46.4%|47.5%|48.3%|48.5%|48.6%|26.0%|27.2%|28.1%|28.4%|28.5%|
> ---
> 2. **Q**: You've examined the use of a uniform sampling procedure as well as one based on the predicted scores but did you also try a voxel-based sampling procedure or similar to maintain the larger distribution of points that, as you discuss in the supplementary material, is necessary for a reliable registration? Could this be the reason why the RR metric is similar with both the 'prob' and 'rand' sampling strategies while the IR is clearly in favor of the 'prob' approach?
>
>     **A**: Though voxel-based sampling is widely adopted in down-sampling points, the output of CoFiNet are correspondences and the two strategies we use are for correspondence sampling.
>
>     The *rand* strategy makes the sampled correspondences uniformly distributed, i.e., sampled correspondences are not too close to each other. As discussed in the Supplementary Material, it is good to the precision of registration. However, as our predicted confidence is not involved, the Inlier Ratio is relatively lower in this strategy. Compared to *rand*, *prob* samples correspondences proportional to predicted confidence and brings improvements on Inlier Ratio. However, as more reliable correspondences tend to lie on more salient regions,  correspondences sampled by the *prob* strategy are usually clustered, which affects the precision of pose estimation.
>
>     Above is one reason why Registration Recall is similar and the other one is RANSAC. Registration Recall is evaluated on relative poses estimated by RANSAC, which is capable of rejecting outliers. Thus, higher Inlier Ratio cannot always promise significant improvements on Registration Recall.
> ---
> 3. **Q**: The results on KITTI seems good, is the RTE and RRE calculated for the correctly registered point cloud or for all?
>
>     **A**: We evaluate our method on KITTI following the settings of D3Feat(CVPR2020) and PREDATOR(CVPR2021). The Registration Recall (RR) is calculated for all the pairs, no matter whether they are correctly registered or not.  The RTE and RRE are only calculated for the correctly registered point clouds, i.e., RTE is only calculated for point clouds whose RTE < $2m$ and RRE is only calculated for point clouds whose RRE < $5^{\circ}$.
> ---
> 4. **Q**: One comment is that the W and the B matrices used in the loss calculations probably should be defined in the paper not only verbally but similarly to what is done in the supplementary. It could also be clearer if the patch-to-patch correspondences are used or the point-to-point correspondences inside of two patches are compared since S~(l) is not defined.
>
>     **A**: Due to the limitation of pages, we present the calculation of W and B as well as the patch-to-patch and point-to-point correspondences in Supplementary Material. Please refer to A.3 Detailed Labels. We definitely consider bringing them back to the main body. Though $\mathbf{\widetilde{S}}^{(l)}$ appears in b.3 of Fig.2, obviously we should define it in the text of the main body. We will fix that in the revised version.
> ---
> 5. **Q**:   Could you clarify the exact nature of the RANSAC procedure, is the GT transformation involved in selecting the correct transformation or is it only based on the number of inliers as would be done in a real-world scenario?
>
>     **A**: The GT transformation is not involved in RANSAC. In our case, rigid transformation is estimated by RANSAC + SVD. RANSAC and SVD are performed for many iterations. For each iteration, a subset of correspondences is sampled and used to solve the transformation by SVD. Then, based on the estimated transformation, all the correspondences can be classified as inliers or outliers. The estimated transformation is later scored as the number of inlier correspondences. After all iterations, the estimated transformation with the highest score will be chosen.

---

### Official Review · Reviewer_ZezU · 2021-07-16

**Rating:** 7
**Confidence:** 2

**Summary:**

The paper presents a novel learning method for point cloud registration by extracting hierarchical correspondences in a coarse to fine manner. There are two modules:

Coarse scale:  a network is first utilized to downsample and extract feature from input  point cloud. The features are then strengthened and used to compute a score matrix for coarse correspondence proposals.

Fine scale: coarse correspondence proposals are expanded to patch correspondences via grouping and refined with a density-adaptive matching module.

**Limitations And Societal Impact:**

- In the coarse scale, how the number of sample points affects the results?
- It would be better to add some examples of failure cases.

**Main Review:**

- It is a novel idea to treat point cloud registration as a coarse-to-fine correspondence problem. The approach is free of keypoint detection and thus keypoint repeatability issue, which enables robust registration.

- The method achieves state-of-the-art results on both indoor and outdoor benchmarks with fewer parameters. The experimental results are detailed.
- The paper is well-written.

**Time Spent Reviewing:**

4

---

> ### Author Response · Authors · 2021-08-10
> **Response to Reviewer ZezU**
>
> We first would like to thank the reviewer for giving us valuable comments.
>
> 1. **Q**:  In the coarse scale, how the number of sample points affects the results?
>
>     **A**: On the coarse level, we sample correspondences by setting a threshold of predicted confidence. In the following table, we experiment with three different thresholds, 0.05, 0.10 and 0.15, among which 0.10 is used in our paper. The #Coarse column in the following table represents the average number of coarse correspondences across all the tested samples. Smaller thresholds result in more correspondences.
>
>     As shown in the following table, when a smaller threshold is used, the Inlier Ratio drops, which means CoFiNet learns reasonable confidence, i.e., there are more outliers when confidence is lower. Larger thresholds bring side-effects to Registration Recall, as they are too strict for some point cloud pairs to propose coarse correspondences. To address the issue, we further propose an adaptive strategy, where we start from a relatively large threshold of confidence (we use 0.2) but adaptively decrease it when the number of proposed correspondences is smaller than another threshold (we use 200). By doing this, we guarantee that each point cloud pair has at least 200 correspondences on the coarse level. This strategy further boosts the performance of CoFiNet on Registration Recall. We will include the new strategy in the revised version.
>
>     **Inlier Ratio**:
>
>     | | | |3DMatch | | | | | |3DLoMatch | | | |
>     |:-:|:-:|:-:|:-:|:-:|:-:|:-:|:-:|:-:|:-:|:-:|:-:|:-:|
>     |#Sampled Points|5000| 2500|1000|500|250|#Coarse|5000| 2500|1000|500|250|#Coarse|
>     |CoFiNet, 0.05|48.3%|49.4%|50.0%|50.2%|50.5%|574.76|25.9%|27.3%|28.2%|28.5%|28.6%|260.34|
>     |CoFiNet, 0.10|50.4%|51.1%|51.6%|51.7%|52.0%|335.34|28.9%|29.8%|30.4%|30.6%|30.8%|127.88|
>     |CoFiNet, 0.15|**54.5%**|**55.5%**|**56.2%**|**56.4%**|**56.6%**|222.18|**32.0%**|**32.7%**|**33.6%**|**33.8%**|**34.0%**|73.89|
>     |CoFiNet, adaptive|49.9%|51.2%|51.9%|52.2%|52.2%|230.07|24.4%|25.9%|26.6%|26.8%|27.0%|203.14|
>
>     **Registration Recall**:
>
>     | | | |3DMatch | | | | | |3DLoMatch | | | |
>     |:-:|:-:|:-:|:-:|:-:|:-:|:-:|:-:|:-:|:-:|:-:|:-:|:-:|
>     |#Sampled Points|5000| 2500|1000|500|250|#Coarse|5000| 2500|1000|500|250|#Coarse|
>     |CoFiNet, 0.05|87.7%|87.4%|87.1%|87%|85%|574.76|62.9%|62.8%|61.9%|60.8%|60.1%|260.34|
>     |CoFiNet, 0.10|87.4%|88.1%|**88.6%**|86.4%|**85.9%**|335.34|63.6%|62.3%|61.8%|60.3%|60.2%|127.88|
>     |CoFiNet, 0.15|84.6%|85.5%|85.1%|85.9%|84.2%|222.18|58.7%|58.9%|58.1%|57.0%|56.5%|73.89|
>     |CoFiNet, adaptive|**89.8%**|**89.5%**|88.4%|**88.3%**|85.7%|230.07|**67.6%**|**66.2%**|**65.8%**|**62.9%**|**61.7%**|203.14|
> ---
> 2. **Q**: It would be better to add some examples of failure cases.
>
>     **A**: We introduce the limitations and discuss some cases on line 84-93 and in Fig. 2 of Supplementary Material. But we do lack some failure cases of registration in Fig. 3 of Supplementary Material. They will be added in the revised version.

---

### Official Review · Reviewer_z7KQ · 2021-07-16

**Rating:** 6
**Confidence:** 1

**Summary:**

The paper proposes an approach for finding correspondences for robust point cloud registration without detecting keypoints. The method is based on the hierarchical coarse-to-fine processing of the point clouds that is then used to find the sought rigid transformation by using RANSAC.

**Ethical Concerns:**

-

**Limitations And Societal Impact:**

-

**Main Review:**

Strengths:
- Keypoints do not have to be detected to find the rigid transformation. This improves the repeatability of the solution.
- The proposed weighting scheme helps in finding the correspondences efficiently.
- The experiments show that the proposed method leads to SOTA results on the 3DMatch and 3DLocMatch datasets. To Table 3, it is not SOTA on Kitti, thus, I suggest removing sentence "Results show that CoFiNet achieves state-of-the-art performance on
14 all evaluated datasets" from the abstract.

Weaknesses:
- The paper (at least for me) was quite challenging to understand. Aside from the topic being far from me, I would suggest significantly improving the text since it is unclear in many places. For example, Fig.1 is supposed to help in understanding the method, however, it does not really help.

Since I am not too familiar with the topic I can't judge if the method is new.
The results seem good compared to recent approaches and, thus, I am suggesting acception.

Suggestion:
- It would be interesting to see the performance if a modern robust estimator is used, e.g. [a,b], not RANSAC.

[a] Raguram, Rahul, et al. "USAC: A universal framework for random sample consensus." IEEE transactions on pattern analysis and machine intelligence 35.8 (2012): 2022-2038.

[b] Barath, Daniel, et al. "MAGSAC++, a fast, reliable and accurate robust estimator." Proceedings of the IEEE/CVF conference on computer vision and pattern recognition. 2020.


**Time Spent Reviewing:**

2

---

> ### Author Response · Authors · 2021-08-10
> **Response to Reviewer z7KQ**
>
> We first would like to thank the reviewer for giving us valuable comments.
>
> 1. **Q**: To Table 3, it is not SOTA on Kitti, thus, I suggest removing sentence "Results show that CoFiNet achieves state-of-the-art performance on 14 all evaluated datasets" from the abstract.
>
>     **A**: On KITTI, we do achieve SOTA performance on Registration Recall. On RRE and RTE, though D3Feat and PREDATOR perform slightly better than CoFiNet, we use only one fourth of their parameters, as shown in Tab. 3. Considering the trade-offs between model size and performance, we call it SOTA in our paper. We will include this suggestion and explain it in the revised abstract.
> ---
> 2. **Q**: The paper (at least for me) was quite challenging to understand. Aside from the topic being far from me, I would suggest significantly improving the text since it is unclear in many places. For example, Fig.1 is supposed to help in understanding the method, however, it does not really help.
>
>     **A**: We definitely consider revising the paper and reformulating the figure to make sure all parts are clear.
> ---
> 3. **Q**: It would be interesting to see the performance if a modern robust estimator is used, e.g. [a,b], not RANSAC.
>
>     **A**: For fair comparison to existing methods, e.g., D3Feat and PREDATOR, the original RANSAC is utilized in our paper. We check the methods suggested, they are originally proposed for estimating homography matrices based on 2D correspondences and haven't provided support for 3D pose estimation. Thus we turn to Graph-Cut RANSAC(CVPR 2018) that currently supports 3D pose estimation. Results can be found in the following table, where we report Registration Recall that is affected by the performance of the robust estimator. We use the *prob* strategy for correspondence sampling. In general, Graph-Cut RANSAC slightly improves the performance.
>
>     **3DMatch**:
>
>     |Sampled Points|5000| 2500|1000|500|250|
>     |:-:|:-:|:-:|:-:|:-:|:-:|
>     |CoFiNet+RANSAC|**87.4%**|88.1%|**86.6%**|86.4%|**85.9%**|
>     |CoFiNet+GC-RANSAC|**87.4%**|**88.4%**|**86.6%**|**87.7%**|85.6%|
>
>     **3DLoMatch**:
>
>     |Sampled Points|5000| 2500|1000|500|250|
>     |:-:|:-:|:-:|:-:|:-:|:-:|
>     |CoFiNet+RANSAC|63.6%|62.3%|61.8%|60.3%|**60.2%**|
>     |CoFiNet+GC-RANSAC|**64.7%**|**64.5%**|**62.0%**|**62.1%**|58.9%|

---

### Official Review · Reviewer_em8a · 2021-07-16

**Rating:** 4
**Confidence:** 5

**Summary:**

This paper proposes a coarse-to-fine network to extract the correspondences for point cloud registration.

**Main Review:**

Originality: The overall framework is similar to PREDATOR, but replace the sparse convolution with KPCov. The attention module follows PREDATOR's transformer. The encoder module is the coarse step and the decoder module is the fine step in this submission. Based on my understanding, the contribution is incremental and the performance achieves very limited improvement compared to PREDATOR.

Quality: This paper is technically sound but I am still worried about the efficiency compared to PREDATOR. The authors have not discussed the weakness of their work. From Figure 3, we can find the final correspondences are located in the very limited regions. Why other regions are all outliers in this work. I am very worried about the robustness since most of the information are discarded roughly in the coarse step.

Clarity: The overall organization is well but could be improved. For example, I disagree with the claim in the abstract "Existing works widely adopt keypoint detection before establishing correspondences but struggle to propose repeatable keypoints". The recent point cloud registration methods, such as pointnetlk(CVPR2019), DGR (CVPR2020), FMR(CVPR2020), PREDATOR (CVPR2021) and PointDSC(CVPR2021), all do not adopt the keypoint detection.

Significance: This work is an increment of PREDATOR by replacing the sparse convolution with KPCov. The significance of the scientific finding is very limited.

**Time Spent Reviewing:**

7

---

> ### Author Response · Authors · 2021-08-10
> **Response to Reviewer em8a**
>
> We first would like to thank the reviewer for giving us valuable comments.
>
> 1. **Q**: The attention module follows PREDATOR's transformer. The encoder module is the coarse step and the decoder module is the fine step in this submission. Based on my understanding, the contribution is incremental and the performance achieves very limited improvement compared to PREDATOR.There are some misunderstandings of our main contributions and superiority in this review.
>
>     **A**: There are some misunderstandings of our main contributions and superiority in this review.
>
>     CoFiNet is clearly different to PREDATOR in many aspects. Most importantly, PREDATOR is a detection-based method that samples keypoints based on their predicted overlap and salient scores, while CoFiNet is a detection-free method, which tackles the correspondence search problem by establishing correspondences consecutively in two scales. On a coarse scale, overlap ratios between pre-computed patches are leveraged as the supervision and node correspondences are extracted by solving an optimal transport problem. On a fine scale, extracted node correspondences are first expanded to patches by grouping and then refined by our proposed CRB with density-adaptive matching modules in Fig.2. The output of PREDATOR is the descriptor and score of each point, while CoFiNet is able to output correspondences directly.
>
>     Concretely, considering computational efficiency, we replace the self-attention module of PREDATOR with the one used in SuperGlue(CVPR2020). Besides, neither our proposed weighting scheme on the coarse scale nor the density-adaptive matching module on the fine scale appears in PREDATOR. Benefiting from the aforementioned designs, CoFiNet avoids keypoint detection and thus keypoint repeatability issues, leading to more robust point cloud registration.  As mentioned by other reviewers and shown in Tab. 1,  CoFiNet gains significant improvements over PREDATOR, especially on 3DLoMatch where point clouds share fewer overlaps.
> ---
> 2. **Q**: This paper is technically sound but I am still worried about the efficiency compared to PREDATOR. The authors have not discussed the weakness of their work.
>
>     **A**:Firstly, compared to existing works, we use much fewer parameters in our model, which can be found in Tab.1 and Tab.3. Besides, we evaluate the runtime of CoFiNet and compare it to that of PREDATOR in Tab. 1 of Supplementary Material, which demonstrates our superiority. Moreover, CoFiNet is able to directly output correspondences, while PREDATOR could only output sampled keypoints and associated descriptors. Thus, PREDATOR's correspondences are established during RANSAC, which is computationally expensive. To demonstrate this point, we further evaluate the runtime of RANSAC. Related results can be found in the following table and the settings follow the experiments in Tab. 1 of Supplementary Material. We will highlight our efficiency in the main body of the revised version.
>
>     |Sampled Points|5000| 2500|1000|500|250|
>     |:-:|:-:|:-:|:-:|:-:|:-:|
>     |PREDATOR| 2.86s|1.25s|0.45s|0.22s|0.11s|
>     |CoFiNet|**0.18s**|**0.11s**|**0.07s**|**0.05s**|**0.05s**|
>
>     For the limitations, we present them in line 84-93 of Supplementary Material and we will either mention them in or move them to the main body in the revised version.
> ---
> 3. **Q**: From Figure 3, we can find the final correspondences are located in the very limited regions. Why other regions are all outliers in this work. I am very worried about the robustness since most of the information are discarded roughly in the coarse step.
>
>     **A**: For Fig.3, in the caption, we mention that this is an extreme case where the overlaps between two point clouds are really limited, which is also illustrated in the leftmost sub-figure of Fig.3. The fact is that correspondences cannot exist on non-overlapped regions. In this case, our proposed correspondences lie on the overlapped areas while PREDATOR produces many outlier correspondences on non-overlapped regions. The last three sub-figures in Fig. 3 demonstrate that our Inlier Ratio is much better than PREDATOR's.
> ---
> 4. **Q**: For example, I disagree with the claim in the abstract "Existing works widely adopt keypoint detection before establishing correspondences but struggle to propose repeatable keypoints". The recent point cloud registration methods, such as pointnetlk(CVPR2019), DGR(CVPR2020), FMR(CVPR2020), PREDATOR (CVPR2021) and PointDSC(CVPR2021), all do not adopt the keypoint detection.
>
>     **A**: Firstly, what we emphasize in the abstract is that,  though many approaches, e.g., USIP(CVPR2019), PRNet(NeuRIPS19), D3Feat (CVPR2020), and the above mentioned PREDATOR(CVPR 2021), are based on and benefit from keypoint detection, proposing repeatable keypoints is still a severe problem for them. We will rephrase this sentence to overcome confusions.
>
>     Moreover, as far as we know, in the provided examples, PointNetLK(CVPR2019) learns a global embedding  for each point cloud and mainly focuses on object-level data with merely thousands of points, while we focus on large-scale scene-level benchmarks. DGR(CVPR2020) and PointDSC(CVPR2021) are proposed to reject outliers and directly estimate the rigid transformation. They take correspondences as input and get rid of the use of RANSAC. They can be combined with any methods that extract correspondences, and for some of them (such as D3Feat and PREDATOR), keypoint detection is included.

---

### Official Review · Reviewer_1Jc9 · 2021-08-02

**Rating:** 5
**Confidence:** 5

**Summary:**

The paper proposed to study the problem of extracting correspondences for 3D point cloud registration. The paper presented CoFiNet (Coarse-to-Fine Network) to extract hierarchical correspondences from coarse to fine without detecting keypoints.  The model firstly learns to match down-sampled nodes and proposes node correspondences. On a finer scale, node proposals are consecutively expanded to patches that consist of groups of points together with associated descriptors. Patch correspondences are then refined to point level by a density-adaptive matching module. The proposed method has been evaluated on both indoor and outdoor standard benchmarks.

**Limitations And Societal Impact:**

The paper did not address the limitations of the work. The paper aims at tackling partial overlap while it did not provide experiments with respect to different ratios of overlapping between input point clouds. It is interesting and critical to analyze the robustness with respect to different ratios of overlapping. The scalability issue has also not been analyzed in the paper. Could the proposed approach be applied to large scale point clouds with low overlapping ratios?

**Main Review:**

Originality: The paper extended the coarse-to-fine strategy in 2D vision tasks to 3D point cloud registration. The related work did not cite and discuss other category of point cloud registration approaches such as the correspondence-free approaches.

Quality: In the paper, the proposed coarse-to-fine pipeline works at a coarse and finer scale. Could this coarse-to-fine strategy be applied to multiple scales?

Clarity: While the paper presented interesting ideas, the clarity is a major concern. For example, the differentiable density-adaptive matching module, which is a key component of the proposed framework, has not been made clear.

Significance: 3d point cloud registration is an important task in computer vision. The paper extended the coarse-to-fine strategy in 2D vision tasks to 3D.

**Time Spent Reviewing:**

5 hours

---

> ### Author Response · Authors · 2021-08-10
> **Response to Reviewer 1Jc9**
>
> We first would like to thank the reviewer for giving us valuable comments.
>
> 1. **Q**: The related work did not cite and discuss other category of point cloud registration approaches such as the correspondence-free approaches. The paper did not address the limitations of the work.
>
>     **A**: As the correspondence-free approaches mainly focus on object-level data and they are similar to methods we mention on line 30-32, we will summarize them together as end-to-end registration methods in the related works. For limitations, due to page limitation, we present them on line 84-93 of Supplementary Material. We will either mention them in or move them to the main body in the revised version.
> ---
> 2. **Q**: Could this coarse-to-fine strategy be applied to multiple scales?
>
>     **A**: Our method can be applied to multiple scales. Suppose that we have $n$ different scales, from coarse (scale 0) to fine (scale n - 1). As the initialization, the CPB in Fig.1 proposes correspondences on the coarsest scale. For scale $i$ $(1 \leq i < n)$, a CRB in Fig.2 can be leveraged to refine correspondences from scale $i - 1$ to the current scale. In our paper, for computational efficiency and brief illustration, we only use a single CRB to refine correspondences directly from the coarsest to the finest scale.
> ---
> 3. **Q**:  For example, the differentiable density-adaptive matching module, which is a key component of the proposed framework, has not been made clear.
>
>     **A**:  As shown in Fig.2, the input of this module is two patches truncated by $k$, which means each patch has exactly $k$ points and associated descriptors. The output of this module is correspondences extracted from the input. As described on line 185-193, some patches could originally have less than $k$ points, where some points will be repeated to fill the empty places. Those repeated points are meaningless and could cause biases during training (see Tab. 2, KITTI).
>
>     In b.1, an all-to-all score matrix is calculated based on descriptors. The slack row and column (blue in figure) are added to deal with points that fail to match. We mute entries calculated from repeated points by setting them to $-\infty $ (red in figure), as the normalization will be performed in the log space. In b.2, we treat correspondence search as an optimal transport problem and solve it via the Sinkhorn algorithm. In each iteration, we drop the slack row and column in row and column normalization, respectively, which guarantees the muted entries always remain unchanged during normalization. In b.3, we use a natural exponential function to obtain the doubly stochastic score matrix, where all the muted entries equal to 0 and thus they are totally ignored when proposing correspondences. We will include some of these details and make sure this part is clear in the revised version.
> ---
> 4. **Q**: The paper aims at tackling partial overlap while it did not provide experiments with respect to different ratios of overlapping between input point clouds.
>
>     **A**: We have conducted experiments with respect to different ratios of overlaps in the main body. In indoor scenarios, we test on both 3DMatch where overlap ratios are larger than 30\% and 3DLoMatch where overlap ratios are between 10\% and 30\%. Please refer to line 222-224, 239-243 for brief introduction of utilized data and Tab. 1 for detailed results. We further compare CoFiNet with PREDATOR in fine-grained overlap ratio intervals in the following table, which shows our superiority over PREDATOR in all the intervals. We use the same settings as the ablation study in Tab. 2. We show Inlier Ratio here, as it is the metric directly affected by the overlap ratios, i.e., lower overlap ratios introduce more ambiguity in matching.
>
>     |Overlap Ratios| 10-20%| 20-30%| 30-40%| 40-50%| 50-60%| 60-70%| 70-80%| 80-90%|90-100%|
>     |:-:|:-:|:-:|:-:|:-:|:-:|:-:|:-:|:-:|:-:|
>     |PREDATOR|9.88%|19.07%|26.34%|33.69%|39.07%|46.24%|52.08%|60.24%|71.46%|
>     |CoFiNet|**20.36%**|**33.81%**|**43.07%**|**50.07%**|**56.43%**|**61.55%**|**64.68%**|**70.46%**|**74.17%**|
> ---
> 5. **Q**:  The scalability issue has also not been analyzed in the paper. Could the proposed approach be applied to large scale point clouds with low overlapping ratios?
>
>     **A**: To prove the scalability of our proposed approach, we conduct experiments on both indoor and outdoor cases. Different from object-level data where each point cloud has merely thousands of points, our utilized benchmarks consist of large-scale scene-level point clouds with more than 10k points.
>
>     For indoor scenarios, we evaluate our method on 3DLoMatch, where point cloud pairs share lower overlaps. For outdoor dataset KITTI where point clouds are larger, generating point clouds with low overlap ratios is impractical, as the ground truth transformation obtained from GPS is too inaccurate for pairs with fewer overlaps. Thus we evaluate our method on standard KITTI following existing works, e.g., PREDATOR(CVPR2021) and D3Feat(CVPR2020).

---

### Decision · Program_Chairs · 2021-09-27

**Decision:**

Accept (Poster)

**Comment:**

This papers introduces CoFiNet, a coarse-to-fine methods to extract hierarchical correspondences between 3D point clouds, which relaxes the need for keypoints detection. Experiments are conducted on datasets with high (3DMatch) and low (3DLoMatch) level of correspondence.

The paper initially received three accept and two reject recommendation. The main concerns pointed out by reviewers were the lack of novelty of the proposed method, and the modest improvement compared to recent baselines (e.g. PREDATOR).
After rebuttal, the initially positive reviewers were convinced by the rebuttal and recommended acceptance. Especially, R5BJR who did a detailed review supported paper acceptance. On the other hand, Rem8a and R1Jc9 still considered the novelty insufficient and the experiments not convincing enough, and stuck on their rejection recommendation.

The AC carefully read the submission and authors' feedback. The AC considers that the idea of adapting keypoint-free coarse-to-fine refinement methods from the 2D to 3D is interesting. Although closely based on prior works, the adaptations and contributions are overall convincing, i.e. the weighting scheme and the differentiable density-adaptive matching. The experiments also show the relevance of the approach compared to recent baselines, e.g. PREDATOR.
The AC thus recommends acceptance, but highly encourages the authors to carefully take into account and reviewers' comments and there feedback to improve the final version of the paper. Especially, the clarity of the paper should be improved to clarify and highlight the contribution and the positioning of the proposed method with respect to related works. The submission has been discussed with the senior area chair who agrees with the recommendation.